# Micellar Hyaluronidase and Spiperone as a Potential Treatment for Pulmonary Fibrosis

**DOI:** 10.3390/ijms22115599

**Published:** 2021-05-25

**Authors:** Evgenii Skurikhin, Pavel Madonov, Olga Pershina, Natalia Ermakova, Angelina Pakhomova, Darius Widera, Edgar Pan, Mariia Zhukova, Lubov Sandrikina, Andrey Artamonov, Alexander Dygai

**Affiliations:** 1Tomsk National Research Medical Centre of the Russian Academy of Sciences, Laboratory of Regenerative Pharmacology, Goldberg ED Research Institute of Pharmacology and Regenerative Medicine, Lenin, 3, 634028 Tomsk, Russia; ovpershina@gmail.com (O.P.); Nejela@mail.ru (N.E.); angelinapakhomova2011@gmail.com (A.P.); artifexpan@gmail.com (E.P.); mashazyk@gmail.com (M.Z.); ermolaeva_la@mail.ru (L.S.); amdygay@gmail.com (A.D.); 2Limited Liability Company «Scientific Future Management», 630559 Novosibirsk, Russia; madonov@scpb.ru (P.M.); artamonov@scpb.ru (A.A.); 3Stem Cell Biology and Regenerative Medicine Group, Whiteknights Campus, School of Pharmacy, University of Reading, Reading RG6 6AP, UK; d.widera@reading.ac.uk; 4Institute of General Pathology and Pathophysiology, 125315 Moscow, Russia

**Keywords:** idiopathic pulmonary fibrosis, hyaluronic acid, electron beam synthesis, poloxamer-hyaluronidase, spiperone, transforming growth factor beta, extracellular matrix

## Abstract

Concentration of hyaluronic acid (HA) in the lungs increases in idiopathic pulmonary fibrosis (IPF). HA is involved in the organization of fibrin, fibronectin, and collagen. HA has been proposed to be a biomarker of fibrosis and a potential target for antifibrotic therapy. Hyaluronidase (HD) breaks down HA into fragments, but is a subject of rapid hydrolysis. A conjugate of poloxamer hyaluronidase (pHD) was prepared using protein immobilization with ionizing radiation. In a model of bleomycin-induced pulmonary fibrosis, pHD decreased the level of tissue IL-1β and TGF-β, prevented the infiltration of the lung parenchyma by CD16^+^ cells, and reduced perivascular and peribronchial inflammation. Simultaneously, a decrease in the concentrations of HA, hydroxyproline, collagen 1, total soluble collagen, and the area of connective tissue in the lungs was observed. The effects of pHD were significantly stronger compared to native HD which can be attributed to the higher stability of pHD. Additional spiperone administration increased the anti-inflammatory and antifibrotic effects of pHD and accelerated the regeneration of the damaged lung. The potentiating effects of spiperone can be explained by the disruption of the dopamine-induced mobilization and migration of fibroblast progenitor cells into the lungs and differentiation of lung mesenchymal stem cells (MSC) into cells of stromal lines. Thus, a combination of pHD and spiperone may represent a promising approach for the treatment of IPF and lung regeneration.

## 1. Introduction

Hyaluronic acid (HA) is of great interest to the scientific and clinical community [1,2]. HA is present in body fluids, tissues, and in the extracellular matrix [3]. In the extracellular matrix, hyaluronic acid performs a structural function by binding to cells and other components of the extracellular matrix through specific and nonspecific interactions. Interacting with proteoglycans, such as aggrecan and versican, HA participates in the organization of fibrin, fibronectin, and collagen [3], and is also involved in the regulation of cell adhesion, migration, and proliferation [1,3,4,5]. 

The course of radiation and bleomycin-induced fibrosis is characterized by a significant increase in the concentration of HA in the lung parenchyma in animals [3,6,7] which is explained by tissue damage (destruction of epithelial cells) and inflammation [8,9]. HA plays an important role in regulating tissue injury and repair and controlling disease outcomes in humans [3]. Thus, hyaluronic acid can act as a marker of fibrotic changes in the lungs. On the other hand, HA is a convenient target for drug therapy.

The main enzyme that regulates the metabolism of HA is hyaluronidase (HD). By cleaving HA into fragments, such as glucosamines and glucuronic acid, the enzyme digests the extracellular matrix [10,11]. Intranasal administration of HD has been shown to reduce the activity fibroblastic processes in the lungs of laboratory animals [6]. In this regard, HD has been proposed to be a potential compound within antifibrotic therapy for idiopathic pulmonary fibrosis.

Due to enzymatic hydrolysis, concentration of HD in the lungs rapidly decreases 2–3 min after administration. This leads to the restoration of the profibrotic viscosity of the hyaluronan matrix and, therefore, to the progression of pulmonary fibrosis. To increase the stability of HD, various forms of pegylated hyaluronidase have been developed [12]. HD conjugated with polyethylene glycol has greater physical stability, lower solubility, reduced sensitivity to proteolytic enzymes, and lower immunogenicity compared to native HD [13,14]. However, the pharmacological activity of pegylated hyaluronidase may be inferior to that of native HD, and the pharmacological effects may not be stable [14]. This could be explained by a change in the structure of the hyaluronidase molecule after conjugation with polyethylene glycol [14,15]. Pluronics, which are block copolymers of polyoxyethylene and polyxypropylene, have been proposed as safe carriers for HD [15]. Under certain conditions, polymer nanoparticles form nanoaggregates (micelles), into which drug molecules can be introduced. The micelle can be composed of a combination of two pluronics or polyethylene glycol with pluronic [16]. Indeed, such structures are have been used for targeted therapy [16,17,18,19,20,21]. We have previously studied the effects of conjugates of pluronics with a predominance of hydrophobic properties and amphiphilic properties with hyaluronidase [22]. Stable antifibrotic effects were mediated by a conjugate based on hydrophobic Pluronic L31 with hyaluronidase (poloxamer-hyaluronidase or pHD).

In the present study, we aimed to evaluate the effects of poloxamer hyaluronidase on bleomycin-induced alveolar damage, inflammation, and lung fibrosis in C57BL/6 mice in comparison with native HD. In addition, we studied the effect of pHD on lung regeneration and on mesenchymal cells as cells involved in fibrosis. Increasing the effectiveness of drug treatment is an important clinical challenge in the treatment of patients with IPF. To enhance the effects of pHD, we supplemented it with a spiperone, which itself has been proven as an anti-fibrotic agent [23].

## 2. Results

### 2.1. Concentration of Hyaluronidase in the Lungs of Mice after a Single Intranasal Administration of HD and pHD in Healthy Mice

After the introduction of native HD (group 1), we detected one phase of absorption of hyaluronidase in the lungs for 1 min of observation, with a maximum concentration of 10 ng/mg. Furthermore, a rapid excretion of the substance from the lungs was observed hyaluronidase in the lungs of mice of group 1 with HD levels of 0.5 ng/mg 20 after application (Figure 1).

After intranasal pHD administration into the lungs, three phases of substance absorption were found (Figure 1). The first phase of hyaluronidase absorption ressembled that of the group of mice which recieved native HD. However, the concentration of the substance in group 2 was three times lower than in group 1. At 5 min, we observed a second phase of absorption in group 2 with 5.13 ng/mL in the lung tissue. This could be explained by a degradation of the least stable fraction of polymerosomes. The third phase is a degradation of the stable fraction of polymerosomes by the reticulo-endothelial system of the lung tissue and a release of hyaluronidase. The maximum concentration of hyaluronidase was reached after 2 h with 7.32 ng/mg. Furthermore, a slow decrease in the concentration of the substance in the lungs was observed within 2–8 h. Eight hours after the administration of pHD, the concentration of hyaluronidase was 1.38 ng/mg.

### 2.2. Concentration of Fibroblastic Process Molecules in Lungs Damaged by Bleomycin during Treatment with pHD Monotherapy and in Combination with Spiperone

Figure 2 shows a graphical scheme of the protocol for inducing pulmonary fibrosis by bleomycin and the protocol for studying the pHD.

Modeling of pneumofibrosis caused an increase in the concentration of HA (by 3.57 times, *p* < 0.05), hydroxyproline (by 3.07 times, *p* < 0.05), type 1 collagen (by 1.99 times, *p* < 0.05), and total soluble collagen (by 1.37 times, *p* < 0.05) in the lung homogenate of group 2 mice compared with mice of intact control (group 1) (Figure 3). However, the concentration of HD1 in group 2 (by 63.2%, *p* < 0.05) was lower than in group 1.

HD treatment had positive effects on hydroxyproline and hyaluronidase 1 levels in the lungs of group 3 mice compared to mice with pneumofibrosis. The effect of pHD administration was superior to that of the HD. As shown in Figure 3, there was a significant decrease in the concentration of hyaluronic acid (by 73.5%, *p* < 0.05), hydroxyproline (by 75%, *p* < 0.05), collagen type 1 (by 54.3%, *p* < 0.05), and total soluble collagen (by 30%, *p* < 0.05) in group 4 compared to mice with pneumofibrosis. After administration of pHD, the level of hyaluronidase in the lung homogenate increased 2-fold (*p* < 0.05) (Figure 3).

The effects of spiperone were similar to those of pHD. Specifically, the hyaluronidase 1 level increased and the level of hyaluronic acid, hydroxyproline, type 1 collagen, and total collagen decreased. We did not observe a potentiation of the effects when spiperone and pHD were co-administred in group 6 (Figure 3).

### 2.3. Concentration of Pulmonary IL-1β, TGF-β, and TNF-α in Bleomycin-damaged Lungs during Monotherapy with pHD and Polytherapy with a Combination of Spiperone and pHD

After bleomycin administration an increase of IL-1β (by 60%, *p* < 0.05), TGF-β (by 35.2%, *p* < 0.05), and TNF-α (by 80.2%, *p* < 0.05) levels was observed in lung homogenates from mice with bleomycin-induced pneumofibrosis compared with intact controls (Figure 4). HD and pHD had no effect on IL-1β and TNF-α, but reduced the levels of TGF-β in group 3 (by 80%, *p* <0.05) and group 4 (by 88.8%, *p* < 0.05) compared to mice with pneumofibrosis. Monotherapy with spiperone and polytherapy with spiperone and pHD significantly decreased the concentration of IL-1β (by 77.3% and 50.2%, respectively) and TGF-β (48.2% and 66%, respectively) (Figure 4).

### 2.4. Histological Parameters of the Lungs Damaged by Bleomycin after Monotherapy with pHD and a Combination of Spiperone and pHD

#### 2.4.1. Effects of HD, pHD, and Spiperone on Tissue Morphology

Staining of lung preparations with hematoxylin and eosin revealed decreased numbers of inflammatory cells, such as lymphocytes, neutrophils, and plasma cells, in the lung parenchyma in mice with pneumofibrosis in group 4 compared to untreated mice from group 2 (Figure 5). A small number of inflammatory cells was observed around the vessels and bronchioles. In group 3 (HD), there was a less significant decrease in the number of inflammatory cells in the lung parenchyma compared to group 4 mice. In this group, the inflammatory infiltrate around the vessels and bronchioles remained at the level of mice with pneumofibrosis. 

Treatment of pulmonary fibrosis with spiperone reduced the infiltration of lymphocytes, neutrophils, and plasma cells into the lungs of group 5 mice compared to untreated mice with pneumofibrosis (Figure 5). In mice treated with spiperone, there was no venous congestion and similar levels of hemorrhages as in animals of group 2.

Sequential administration of spiperone and pHD reduced the activity of the inflammatory response in the lungs of mice in group 6 more efficiently compared with monotherapy in groups 4 and 5 (Figure 5). Polytherapy led to a complete resolution of the inflammatory reaction in the lung parenchyma around the vessels and bronchioles. Moreover, no venous congestion and hemorrhages were observed. 

#### 2.4.2. Evaluation of Collagen Fibers

Van Gieson staining of lung preparations with picrofuchsin revealed a significant decrease in the area of connective tissue (by 52%, *p* < 0.05) in the lungs of mice in group 4 after administration of pHD compared to mice in group 2 (Figure 6, Appendix A). In contrast, in group 2, native HD had no effect on connective tissue in the lungs.

Spiperone prevented the deposition of connective tissue in the lungs of mice in group 5 compared to mice in group 2 (decreased of 48.8%, *p* < 0.05) (Figure 6, Appendix A). Polytherapy with spiperone and pHD resulted in a significantly higher decrease of the area of connective tissue in mice of group 6 compared to animals that underwent monotherapies with spiperone (group 5) and pHD (group 4). The area of connective tissue in group 6 was 32.5% (*p* < 0.05) relative to group of mice with pneumofibrosis.

### 2.5. Influence of Spiperone and pHD on the Expression of CD16, CD31, and Pan-Cytokeratin in Bleomycin-Damaged Lungs

Immunohistochemical analysis of lung preparations revealed a decrease in the expression of the inflammatory cell marker CD16 in the lungs of mice with pneumofibrosis when treated with spiperone (by 60%, *p* < 0.05) and pHD (by 44%, *p* < 0.05) compared with untreated mice of group 2 (Figure 7). The effects of spiperone and pHD were extended to predominantly diffuse expression of CD16. The effects on perivascular and peribronchial marker expression were negligible.

Spiperone increased the expression of CD31 (by 64.4%, *p* < 0.05) and pan-cytokeratin (by 18%, *p* < 0.05) in the lung tissue of group 4 mice relative to group 2 mice. The effects of spiperone increased the expression of CD31 in the alveoli and the lumen of blood vessels. pHD had no effect on CD31^+^ cells and pan-cytokeratin^+^ cells in group 4 (Figure 7).

### 2.6. Cytometric Evaluation of Mesenchymal Stem Cells in Bleomycin-Damaged Lungs during Treatment with pHD and Spiperone

Bleomycin injection led to an increase of the number of mesenchymal stem cells (MSC) in the lungs of mice with pneumofibrosis (by 51%, *p* < 0.05) compared to intact contol mice (Figure 8). HD had no effect on MSCs in group 3. After administration of pHD, the number of MSCs in group 4 (by 80%, *p* < 0.05) was higher than in group 2. Spiperone monotherapy and polytherapy with spiperone and pHD, on the contrary, reduced the number of MSCs in the lungs of mice in group 5 (by 37%, *p* < 0.05) and group 6 (by 39.2%, *p* < 0.05) compared with untreated mice with pneumofibrosis.

### 2.7. Cell Culture Studies

#### 2.7.1. Influence of Spiperone, HD, and pHD on CFU-F

As shown in Figure 9, the clonal activity of fibroblast progenitor cells increased significantly, primarily in the culture of bone marrow CD45 cells (d3, d7). At later stages, an increase in the number of CFU-F was observed in the culture of blood CD45^-^ cells (d7, d21) and lung CD45^-^ cells (d7, d14, d21).

The main effects of the drugs were observed on d14 and d21. Thus, pHD did not affect the clonal activity of fibroblast progenitor cells in the bone marrow and blood. On the other hand, treatment reduced the number of CFU-F (by 56%, *p* < 0.05 compared with group of mice with pneumofibrosis) in the lung cell culture on d21 (Figure 9).

Spiperone had a suppressive effect on the growth of CFU-F in the culture of fibroblast progenitor cells isolated from bone marrow (d3, d21), blood (d21), and lung (d7, d14) of group 5 compared with group of mice with pneumofibrosis (Figure 9).

Fibroblast progenitor cells isolated from the bone marrow and blood of group 6 mice on d14 and d21 generated the same number of colonies as the corresponding cells in group 4 mice (Figure 9). However, the clonal activity of fibroblast progenitor cells of the lungs of group 6 was significantly lower than in group 2 on the d14 and d21, and was comparable with group 5 on the d14 and with group 4 on the d21.

#### 2.7.2. Multilineage Differentiation of MSCs

Pulmonary fibrosis did not affect the chondrogenic and adipogenic differentiation of MSCs in group of mice with pneumofibrosis (group 2) compared with group 1 at the time of the highest content of connective tissue in the lungs (d21) (Appendix A). At the same time, we observed a decrease (by 78%, *p* < 0.05) in the area of mineralized areas (osteogenic differentiation) (Appendix A) and an increase (by 52%, *p* < 0.05) in the number of fibroblasts in the corresponding cultures of group 2 (Figure 10).

After spiperone treatment, the number of mesenchymal cells with lipophilic inclusions in group 5 was significantly lower (by 72%, *p* < 0.05) than in group 2 (Appendix A). In addition, relative to group 2, spiperone had an inhibitory effect on adipogenic, chondrogenic, and fibroblastic differentiation of MSCs. In the first case, in vitro, the number of cells containing lipid inclusions decreased by 63.7% (*p* < 0.05), in the second case, the number of cells containing sulfated proteoglycans decreased by 34% (*p* < 0.05), in the third case, the content of fibroblasts decreased by 36.7% (*p* < 0.05) (Figure 10).

## 3. Discussion

IPF is a chronic progressive disease. IPF is characterized by a development of interstitial fibrosis and progressive respiratory failure [24,25]. IPF prognosis is poor and the life expectancy after diagnosis is five years [26,27]. The lack of effective treatment in patients with IPF predetermined the present study. Hyaluronidase has been proposed as a molecule with potential therapeutic activity. HD is the main enzyme that regulates the metabolism of hyaluronic acid. By cleaving HA into glucosamines and glucuronic acid, HD remodulates the extracellular matrix [10,11]. In experimental pulmonary fibrosis, hyaluronidase reduces the concentration of glucuronic acid and thus prevents the development of fibrosis [6]. However, due to enzymatic hydrolysis, the concentration of HD in the lungs rapidly decreases. This leads to the restoration of the profibrotic viscosity of the hyaluronan matrix and the progression of pulmonary fibrosis.

To deliver a pharmacologically active molecule to the target and prevent its hydrolysis in an aggressive environment, the use of Pluronic has been suggested [14,16]. Similar structures are used for targeted therapy [14,16,19,20,21]. We have studied the conjugate of Pluronic L31 and hyaluronidase 1 as a potential therapeutic agent for the treatment of pulmonary fibrosis. With the help of electron beam exposure, a polymer micelle with an incorporated enzyme was formed.

In the first part of this study, we revealed a significantly greater stability of pHD in comparison with native HD with intranasal administration. Five minutes after intranasal administration of HD, traces of the enzyme were found in the lungs. After intranasal administration of pHD, three phases of degradation were observed (Figure 1). The third phase of the degradation of pHD was the longest: within 2–8 h, a slow decrease in the concentration of the substance in the lungs was observed. The maximum value of the concentration of hyaluronidase was reached after 2 h of observation.

The second part of the study is devoted to the study of the effects of pHD in comparison with HD in a model of pulmonary fibrosis on the 21st day of the experiment. In mice treated with pHD (group 4), a complete resolution of the inflammatory response in the lung tissue was observed with only a small number of inflammatory cells present around the vessels and bronchioles (Figure 5). The HD administration group of mice retained a larger number of inflammatory cells in the lung parenchyma compared to mice treated with pHD, peribronchial, and perivascular inflammation persisted (Appendix A). pHD significantly reduced the fibrotic response to bleomycin injury of the alveolar epithelium. The content of connective tissue in the lungs decreased by more than 2-fold in relation to group of mice with bleomycin-induced pneumofibrosis (Figure 6). In addition, therapy with pHD decreased the concentrations of TGF-β, hydroxyproline, collagen type 1, and total collagen in the lung homogenate. Moreover, it induced a sharp decrease in the ratio of hyaluronic acid/hyaluronidase 1 to 0.24 at 0.33 in group of intact control and 3.24 in group of mice with pneumofibrosis. HD did not significantly affect most of these parameters. Additionally, immunohistochemistry was carried out to assess the regeneration of epithelial and endothelial cells. We did not observe significant changes in the expression of CD31 and pan-cytokeratin in the lungs of mice in group 4 relative to group of mice with bleomycin-induced pneumofibrosis.

According to previous reports, polymersomes are adsorbed on the basal and cell membranes [17]. However, changes in the physicochemical properties of membranes, such as viscosity and permeability after which the degradation of the polymerosome occur [28]. The pHD conjugates studied in this work are about 100 nm in size. Particles of this size are called the “respirable fraction” and have the maximum pharmacological effect, since they penetrate without destruction into the alveoli, where, after degradation of the polymer carrier, hyaluronidase re-modulates the hyaluronan matrix.

We have previously demonstrated the anti-inflammatory and anti-fibrotic effects of spiperone [23]. We hypothesized that co-administration of spiperone and pHD could have higher therapeutic efficacy compared to a monotherapy with each respective compound. 

According to our data, spiperone reduces the edema of the alveolar epithelium, exudation, and infiltration of the lung parenchyma, walls, and lumen of the alveoli by inflammatory cells, including CD16^+^ cells, in group 5 compared with mice in group 2 (d21). Simultaneously, a decrease in the degree of desquamation of alveocytes in the lumen of the alveoli has been observed. As a result, no obliteration of the alveoli occurs, and the destruction of lung tissue slows down. In addition, the lung preparations of group 5 lack cysts, which were typical for group 2 (Figure 5). The area of connective tissue in the lungs of mice treated with spiperone was 50% (*p* < 0.05) lower than in untreated mice. In confirmation of the anti-inflammatory and antifibrotic activity of spiperone, ELISA data indicated a decrease in the expression of inflammation and fibrosis factors (IL-1β, TGF-β) and fibrogenesis molecules, such as hydroxyproline, type 1 collagen, total collagen, in the lungs. As mentioned earlier, destruction of the profibrotic hyaluronic matrix is particularly important for anti-fibrotic therapy. Spiperone shifted the ratio of hyaluronic acid/hyaluronidase 1 towards the enzyme: in group 5 this ratio is 0.71, at 3.24 in group 2, and 0.33 in group 1. This is noteworthy, since dopamine control is assumed to at least partly control the metabolism of hyaluronic acid and hyaluronidase. However, an interaction of the D2 signaling pathway with hyaluronidase 1 and hyaluronic acid in experimental fibrosis has not been described before.

In lung fibrosis, restoration of damaged tissues and cells is one of the important criteria for the effectiveness of the treatment. According to our data, spiperone increased the expression of epithelial markers (pan-cytokeratin) and endothelium (CD31) in the lung parenchyma of group 5 mice compared to group 2 (Figure 7). These results indicate regeneration of bleomycin-damaged lung tissue. In this regard, implication of blockade of dopamine D2 receptors in IPF therapy is significantly increasing.

Since mesenchymal cells participate to the development of pulmonary fibrosis [29,30,31], we assessed the effects of spiperone on these cell populations. Spiperone significantly reduced the population of MSCs in the lungs of mice in group 5 compared with group 2 on the 21st day of the experiment (Figure 9). At the same time, a decrease in the activity of MSC differentiation into adipocytes, osteoblasts, chondrocytes, and fibrocytes was observed. Notably, cells such as fibrocytes produce collagen [31]. Another important aspect of the action of spiperone is associated with a decrease in the clonal activity of fibroblast progenitor cells in the following chronological order: in the bone marrow, blood, lungs (Figure 10 and Appendix A). All these data suggest dopamine regulation of various functions, such as mobilization, migration, proliferation, and differentiation, of mesenchymal cells and inhibition of these processes by spiperone.

Next, we studied the effects of sequential administration of spiperone and pHD (polytherapy) in the simulation of pulmonary fibrosis (d21). Polytherapy prevented the infiltration of the parenchyma and alveoli by inflammatory cells (lymphocytes, neutrophils, plasma cells), a significantly smaller number of inflammatory cells were localized around the vessels and bronchioles than with spiperone monotherapy and pHD (Figure 5). In the walls of the alveoli in animals of group of mice with pneumofibrosis and treated spiperone and pHD, there was no venous plethora and hemorrhages. These data can be interpreted as an absence of inflammatory reaction in the lung tissue. In group 6, the decrease in the level of tissue IL-1β was greater than in group 4, but less prominent than in group 5 (Figure 4). However, an opposite pattern was observed for TGF-β.

Polytherapy inhibited the synthesis and deposition of collagen in the bleomycin-damaged lungs more intensely than monotherapy with pHD or spiperone. Figure 3 shows that the sequential administration of drugs reduced the concentration of fibrogenesis molecules. In mice with pulmonary fibrosis treated with spiperone and pHD, the content of connective tissue did not significantly differ from that in group of mice from intact control, and the value of the ratio hyaluronic acid/hyaluronidase 1 decreased to 0.43. We did not observe similar effect after a monotherapy.

The data presented in this study indicate that prescribing D2 dopamine receptor antagonists the subsequent destruction of the profibrotic matrix of hyaluronan by poloxamer hyaluronidase resistant to the action of enzymatic factors might represent a new therapeutic avenue to treatment of IPF. Notably, the proposed approach could lead to a regeneration of damaged and lost endothelial and epithelial cells in patients with IPF.

## 4. Materials and Methods

### 4.1. Animals

Nine-week-old male C57BL/6 mice (Surgical Bio-modelling Department of the Goldberg ED Research Institute of Pharmacology and Regenerative Medicine, Tomsk, Russia) were used in all experiments. Animals were randomly assigned into the experimental groups. All experimental protocols were approved by the animal care and use committee of the Goldberg ED Research Institute of Pharmacology and Regenerative Medicine, Tomsk NRMC (IACUC Protocol No. 94092015, 17.09.2015). Within this study, 210 mice were used.

### 4.2. Modeling of Experimental Pulmonary Fibrosis

Experimental pulmonary fibrosis was induced by a single intratracheal bleomycin (BLM, Nippon Kayaku Co., Ltd., Tokyo, Japan) administration at a dose 80 μg/mouse in 0.03 mL of 0.9% NaCl, which was slowly instilled in the tracheal lumen [32]. All procedures were performed under anesthesia induced by inhalation of isoflurane using an apparatus for inhalation anesthesia UGO BASILE model 21050 (UGO BASILE, Comerio, Italy). These animals formed the BLM control. Control animals were administered a single intratracheal 0.03 mL 0.9% NaCl. The introduction of BLM was defined as day zero (d0). All mice were euthanized on d21 by CO_2_.

Mice were cohoused (five to six mice per cage) and entrained to a reverse 12 h light/12 h dark cycle. Throughout the experimental period, mice had ad libitum access to standard rodent chow.

### 4.3. Reagents

#### 4.3.1. Hyaluronidase

Bull testicular hyaluronidase (HD) was kindly provided for research by LLC Scientific Future Management (Novosibirsk, Russia). In stage 1 of the experiment, HD (40 U/kg) in a volume 8 μL buffer/mouse was administered intranasally once. During the 2nd and 3rd stages of the experiment, HD was administered intranasally at a dose of 40 U/kg in a volume of 8 μL buffer/mouse once a day on days 10–18 of the experiment.

#### 4.3.2. Poloxamer Hyaluronidase

The drug for the study is a conjugate of a polymer carrier Pluronic L31 (predominance of hydrophobic properties) and testicular hyaluronidase (hyaluronidase 1, provided for research by LLC Scientific Future Management (Novosibirsk, Russia): poloxamer-hyaluronidase (pHD). During the 1st stage of the study, mice were injected with pHD at a dose of 40 U/kg intranasally in a volume of 8 μL/mouse PBS once. During the 2nd and 3rd stages of the study, mice were received pHD at a dose of 40 U/kg intranasally in a volume of 8 μL/mouse PBS once a day on days 10–18 of the experiment.

#### 4.3.3. Spiperone

Selective, competitive antagonist of dopamine D_2_-receptors spiperone was obtained from Sigma-Aldrich (Sigma-Aldrich, St. Louis, MO, USA). During monotherapy, spiperone was injected intraperitoneally at a dose of 1.5 mg/kg in 100 μL of physiological saline once a day on days 10–18 of the experiment. During polytherapy, spiperone was injected intraperitoneally at a dose of 1.5 mg/kg in 100 μL of physiological saline once a day on days 10–18 of the experiment, 1 hour before the pHD introduction.

### 4.4. Study Design and Experimental Groups

At the first stage of the study, the concentration of hyaluronidase in the lung homogenate in mice after a single intranasal administration of HD and pHD was evaluated using ELISA. The mice were allocated into two groups (*n* = 45/group): group 1 (mice treated with HD; group 2 (mice treated with pHD). Blood samples were taken 1 h before administration (0 min) and after 1, 5, 10, 20, and 40 min and after 1; 2; 4; and 8 h after the introduction of HD and pHD (Figure 11).

The second stage of the study was devoted to the study of the effects of native hyaluronidase and pHD on a bleomycin-induced pneumofibrosis model (Figure 2). In addition, the possibility of enhancing the effects of pHD with spiperone has been studied. The mice were allocated into 6 equal groups (*n* = 10/group): intact control is group 1, which is mice without bleomycin and those treated with the solvent made up the control group; pneumofibrosis is in group 2, which is mice receiving bleomycin; pneumofibrosis+HD are in group 3, which is pneumofibrosis, treated with HD; pneumofibrosis + pHD are in group 4, which is pneumofibrosis treated with pHD; pneumofibrosis + spiperone are in group 5, which is pneumofibrosis treated with spiperone; pneumofibrosis + spiperone + pHD are in group 6, which is pneumofibrosis treated with spiperone and pHD. On the 21st day of the experiment, inflammation, connective tissue, and lung damage were histologically evaluated in mice of groups 1–6; ELISA levels of IL-1β, TGF-β, TNF-α, type I collagen, hydroxyproline, and hyaluronic acid in the lung homogenate.

During the third stage, the effect of spiperone and pHD on MSCs and fibroblast progenitor cells was studied using a pneumofibrosis model. The mice were allocated into 5 groups (*n* = 10/group): intact control is group 1, which is mice without bleomycin and those treated with solvent made up the control group; pulmonary fibrosis is group 2, which is mice receiving bleomycin; pulmonary fibrosis + pHD are in group 4, which is pulmonary fibrosis, treated with pHD; pulmonary fibrosis + spiperone are in group 5, which is pulmonary fibrosis treated with spiperone; pneumofibrosis + spiperone + pHD are in group 6, which is pulmonary fibrosis treated with spiperone and pHD. On the 21st day of the experiment, the content of lung MSCs, the content and clonal activity of fibroblast progenitor cells of the lungs were assessed by cytometric and cultural methods in mice of groups 1, 2, 4–6, the differentiation of lung MSCs was studied in mice of groups 1, 2, and 5 (Figure 12).

### 4.5. Enzyme-Linked Immunosorbent Assay

#### 4.5.1. Hyaluronate-endo-β-*N*-acetylhexosaminidase Measurement

The hyaluronoglucosaminidase 1 (HYAL1) or hyaluronidase (HD) level in lung homogenate of right lung lobes was determined by ELISA according to manufacturer instructions (Bovine Hyaluronidase-1 (HYAL1) ELISA kit, MBS7238849, MyBioSource, Inc., San Diego, CA, USA). The right lung lobes were excised and snap frozen after having measured the wet weight. Sensitivity was >0.1 ng/mL.

#### 4.5.2. IL-1β, TGF-β, TNF-α Measurements

The concentrations of IL-1β, TNF-α, and TGF-β in lung homogenate of right lung lobes were determined by ELISA according to manufacturer instructions (Cusabio Biotech CO., Ltd., Wuhan, China). Sensitivities were >7.8 pg/mL for IL-1β, >4.7 pg/mL for TNF-α, and >0.2 ng/mL for TGF-β. 

#### 4.5.3. Hyaluronic Acid, Hydroxyproline, Collagen Type I, and Total Soluble Collagen Measurements

Hyaluronic acid, hydroxyproline, and collagen type I levels were determined by ELISA according to the manufacturer instructions (Cusabio Biotech CO., Ltd, Wuhan, China). Right lobes of the lung were isolated, weighed, and frozen immediately. Sensitivities were >15.6 pg/mL for hyaluronic acid, >1.95 ng/mL for hydroxyproline, >0.039 ng/mL for collagen type I. Results were expressed as ng (pg) per right lung.

The total soluble collagen was determined using a standard curve for the SircolTM assay (S1000, Biocolor Ltd., Country Antrim, UK) according to the manufacturer’s instructions. The right lung lobes homogenate supernatants were placed in 1.5 mL tubes. Sircol-dye was added, the content of the tubes homogenized for 30 min and centrifuged for 10 min (10,000× *g*). The pellets were dissolved with alkaline reagent. Absorbance was read at 540 nm. Detection Limit: 1.0 µg. The amount of total soluble collagen per whole right lung was measurement. Results were expressed as µg collagen per mL. 

### 4.6. Histological Examination of Lung Tissue

#### 4.6.1. Analysis of Lung Tissue

The morphological examination of lungs was performed on d21. For histological studies, left lung lobes were fixed in 4% neutral formalin solution, embedded in paraffin, and sections with thickness of 5 µm were prepared. Subsequently, sections were staining with hematoxylin and eosin. Histological examination was carried out in three areas of lung tissue (upper, middle, and lower) on d21 as described previously [33]. Lung structure, presence of edema, infiltration by proinflammatory cells as well as venous hyperemia, and vascular and bronchial walls thickening were assessed [34,35,36]. Micropreparations from each experimental animal were examined under the light microscope Axio Lab.A1 (Carl Zeiss, MicroImaging GmbH; Göttingen, Germany) at 100× and 400× magnifications.

#### 4.6.2. Analysis of Pulmonary Inflammation

The degrees of peribronchial and perivascular infiltrates were assessed by the scale of inflammation and quantitative assessment of peribronchial and perivascular mononuclear cells [37]. The drugs were coded and peribronchial inflammation was assessed in a blinded manner using a reproducible scoring system. Each tissue slice was assigned a value from 0 to 3. A value of 0 was taken when no inflammation was detected, a value of 1 was when there was a rare encounter with inflammatory cells, a value of 2 was when most of the bronchi or vessels were surrounded by a thin layer (from one to five cells) of inflammatory cells, and a value of 3 was when most of the bronchi and blood vessels were surrounded by a thick layer (more than five cells) of inflammatory cells. Since there were five to seven randomly selected tissue sections per mouse, inflammation scores could be expressed as an average per animal and compared between groups.

In addition, cells in the peribronchial and perivascular segments were counted relative to the length of the basement membrane. The total index of bronchial and vascular inflammation was expressed as the number of inflammatory cells/μm of the basement membrane. 

#### 4.6.3. Analysis of Pulmonary Fibrosis

To count collagen fibers in lung parenchyma, the histological slides were stained by picrofuchsin using the Van Gieson method [38,39]. At least 10 photomicrographs without overlapping across the cut surface of the lung tissue at 100× magnification were taken for each experimental animal. The used system consisted of a microscope (Axio Lab.A1, Carl Zeiss MicroImaging GmbH; Göttingen, Germany) with a video camera (AxioCam ERc5s, Carl Zeiss; Göttingen, Germany), connected to a personal computer. Gathered images were processed using the software AxioVision Rel.4.8.2. The content of collagen fibers in lung was determined using a function for counting the area of the object in the image. Bronchovascular strands were carefully removed from the analyzed areas. The count of collagen fibers (X) was calculated using the formula:X = Σ a × 100/(S − Σ b), (1)
where ∑ a is the sum of the pixels occupied by tissue with fibrosis in 10 images of one specimen, S is the number of pixels corresponding to the full area of the image (when using this camera and the program—4423680), b is the sum of pixels occupied by the empty part of the slide, in 10 images one drug [38,39]. Content of the connective tissue in the lungs was expressed as a percentage of the lung tissue area.

### 4.7. Immunohistochemical Examination of the Lungs

Immunohistochemical examination of the lungs was performed on d21. Sections of lung tissue were placed on glass slides with an adhesive polylysin coating (Leica BioSystems, Wetzlar, Germany). Before staining, tissue sections were dewaxed with the following antigen unmasking in a citrate buffer (pH = 6) for 20 min. Incubation with primary antibodies was performed in a wet chamber at 37 °C. The following primary markers were used to identify specific cell markers antibodies: polyclonal antibodies to membrane protein CD31 (ab28364, Abcam, Cambridge, MA, USA), monoclonal antibodies to CD16 (ab198169, Cambridge, MA, USA), monoclonal antibodies to pan-cytokeratin (AE1/AE3) (ab80826, Abcam, Cambridge, MA, USA). For antibody detection, an imaging system was used in accordance with the manufacturer’s instructions (Spring Bioscience, Pleasanton, CA, USA). The contrast of the sections was performed with hematoxylin. After staining, the slices were dehydrated in xylene and enclosed in the installer environment. The microscope Axio Lab.A1 (Carl Zeiss, MicroImaging GmbH; Göttingen, Germany) with the AxioCam ERc5s camera (Carl Zeiss, Göttingen, Germany) was used to obtain micrographs. The specimens were scanned randomly in their entirety without choosing the same field more than once, by the sequential movement of the mechanical stage (10 micrographs for one slice). The analysis of the obtained images and the counting of cells expressing detectable antigens were performed using the ImageJ program (Madison, WI, USA), which quantified the number of positive cells at 100× magnification [40,41]. 

### 4.8. Flow Cytometric Analysis

Mononuclear cells from lungs were isolated as described earlier [42] and the expression of surface markers on mononuclear cells derived from lungs was analyzed. Fc-receptors were blocked by pre-incubation of the cells with unconjugated anti-CD16/CD32 antibodies for 10 min (eBioscience, San Diego, CA, USA, Clone: 93, Cat# 14-0161-85, 1/50 dilution) in 50 μL of 0.1% saponin (Sigma-Aldrich, St. Louis, MO, USA, Cat# S4521) and 1% BSA (Sigma-Aldrich, St. Louis, MO, USA, Cat# A3059-100G) in phosphate buffered saline (PBS) per tube. After the pre-incubation, cells suspensions were stained with fluorophore-conjugated monoclonal antibodies: CD45 PerCP (QC Testing: Mouse, Clone: 30-F11, Cat# 557235, 1/100 dilution) or CD45 APC-Cy 7 (QC Testing: Mouse, Clone: 30-F11, Cat# 557659, 1/100 dilution), CD31 APC (QC Testing: Mouse, Clone: MEC 13.3, Cat# 551262, 1/50 dilution), CD34 FITC (QC Testing: Mouse, Clone: RAM34, Cat# 560238, 1/50 dilution), CD73 V450 (QC Testing: Mouse, Clone: TY/23, Cat# 561544, 1/50), CD44 APC (QC Testing: Mouse, Clone: IM7, Cat# 559250, 1/50), CD90 PerCP (QC Testing: Mouse, Clone: OX-7, Cat# 557266, 1/50), CD106 (VCAM-1) FITC (QC Testing: Mouse, Clone: 429 (MVCAM.A), Cat# 561678, 1/50) (all Becton Dickinson, San Jose, CA, USA). All antibodies were titrated to determine their optimal staining concentration and appropriate isotype controls were used. Labeled cells were washed thoroughly with 500 μL of FACSFlow (Becton Dickenson, Franklin Lakes, NJ, USA, Cat# 342003).

All samples were run on a Becton Dickenson FACSCanto II flow cytometer. The instrument was set up and standardized using BD Cytometer Setup and Tracking (CS&T) procedures according to manufacturer specifications. One hundred thousand events per tube were acquired. Data were analyzed using FACSDiva 8.0 software.

Thus, mesenchymal stem cells were characterized as CD45^‒^ CD106^+^CD44^+^CD73^+^CD90^+^, fibroblast progenitor cells as CD45^‒^CD31^‒^CD34^+^CD90^+^.

### 4.9. In Vitro Study of MSCs and Fibroblast Progenitor Cells of the Lungs

#### 4.9.1. Isolation of Lung Cells

The chest cavity was surgically opened under sterile conditions. All metal instruments were sterilized by autoclaving; sterile disposables were from BD Falcon. Lungs were isolated from surrounding tissue, rinsed with phosphate buffered saline (PBS, Sigma-Aldrich, St. Louis, MO, USA), mechanically dispersed, and minced with scissors. Blood was washed out of the lungs by perfusion of the right ventricle of the heart using 3–5 mL of chilled Hanks’s phosphate buffer (HBSS, Sigma-Aldrich, St. Louis, MO, USA). In a Petri dish with a diameter of 90 mm with HBSS, the trachea and large bronchi were removed. Next, the tissue was minced using disposable scalpels, with a sufficient volume of liquid, and treated with a 0.2% collagenase II type (Sigma-Aldrich, St. Louis, MO, USA) solution, diluted with sterile HBSS, preheated to 37 °C. The mixture was then incubated at 37 °C for 30 min. To remove the stroma and large aggregates, the cell suspension was filtered through a 70 μm nylon cell filter (BD Falcon, Franklin Lakes, NJ, USA). The cell suspension was centrifuged for 10 min at 300× *g* and the supernatant was discarded. Carefully the precipitation was resuspended, using 2 mL of cold medium of the following composition: 90% DMEM-LG, 10% inactivated fetal bovine serum (FBS), 2 mM l-glutamine, antibiotic solution (penicillin/streptomycin 100 U/mL and 100 μg/mL) (All Sigma-Aldrich, St. Louis, MO, USA). The resulting aspirate was transferred to a test tube with an equal volume of HBSS with 2% inactivated FBS. The resulting suspension was layered on a Histopaque-1077 density gradient in a 2:1 suspension-gradient ratio and centrifuged for 25 min at 300× *g*. The cells of the interphase ring were harvested and diluted with a 5-fold volume of HBSS with 2% inactivated FBS and washed twice by centrifugation at 250× *g* for 7–10 min. Cells were counted in hemocytometer [42,43].

#### 4.9.2. Cultivation of MSCs

Lung mononuclear cells in an amount of 0.5–1.0 × 10^8^ were resuspended in 7 mL of culture medium warmed to 37 °C, consisting of 90% DMEM-LG (Sigma-Aldrich, St. Louis, MO, USA), 10% inactivated FBS (Sigma-Aldrich, St. Louis, MO, USA), 2 mM l-glutamine (Sigma-Aldrich, St. Louis, MO, USA), antibiotic solution (penicillin/streptomycin 100 U/mL and 100 μg/mL; Sigma-Aldrich, St. Louis, MO, USA). Then, the cell suspension was introduced into a culture flask with a vented cap and an area of 25 cm^2^ (Techno Plastic Products AG, Trasadingen, Switzerland) and incubated at 37 °C, 5% CO_2_, and 100% air humidity for 72 h. Then, the supernatant with non-adherent cells was removed, and the base culture medium was replaced with a new portion. Preliminarily, markers of MSCs were studied in some of the adherent cells. Subsequent changes of the medium were made every 3–4 days until 70–90% coverage of the vial surface with cells was achieved, depending on the group.

After reaching the maximum confluence (70–90% of the total area of the flask), the cell culture was washed twice with phosphate buffer (Sigma-Aldrich, St. Louis, MO, USA). The cells were removed from the plastic surface using a 13 mm wide silicone spatula (Techno Plastic Products AG, Trasadingen, Switzerland). The removed cells were washed off with PBS (Sigma-Aldrich, St. Louis, MO, USA) for subsequent use (second passage). The concentration of mononuclear cells during repeated passaging was 0.5–1.0 × 10^7^ in 7 mL of the base culture medium. At the end of the cultivation cycle, the cell culture was washed twice with phosphate buffer in a culture flask and removed from the surface with a silicone spatula. A suspension of mononuclear cells was transferred into test tubes and centrifuged at 300× *g* for 5–7 min. The supernatant was replaced with 1–2 mL of the basic culture medium, the number of mononuclear cells was counted and the immune profile of MSCs was assessed.

At each change of the base medium and upon reaching the maximum confluence, a morphological assessment of the state of the cell culture was carried out. The doubling time (tD) of endothelial and epithelial cells, and fibroblast cells was calculated by the formula:tD = (log2 × t )/(logN_h_ − logN_0_) (2)
where N_0_ is the number of cells during the second change of the medium, N_h_ is the number of cells during the studied period of cultivation t [42,43].

#### 4.9.3. Differentiation of Lung MSCs In Vitro

To confirm osteogenic differentiation, cells obtained after long-term cultivation were transferred into a 6-well culture plate (2.5 × 10^5^ cells/well) in a culture medium containing 90% DMEM-LG (Sigma-Aldrich, St. Louis, MO, USA), 10% FBS (Sigma-Aldrich, St. Louis, MO, USA), and supplemented with 0.2 mM l-ascorbic acid 2-phosphate (Sigma-Aldrich, St. Louis, MO, USA), 10 mM β-glycerophosphate (Sigma-Aldrich, St. Louis, MO, USA), 1 × 10^−8^ M dexamethasone (Sigma-Aldrich, St. Louis, MO, USA), 100 U/mL penicillin, and 100 μg/mL streptomycin (Sigma-Aldrich, St. Louis, MO, USA). The medium was changed every 3–4 days. On day 21, the cell culture was washed twice with phosphate buffer and fixed in 4% formaldehyde (Sigma-Aldrich, St. Louis, MO, USA) for 10 min. Mineralization of the extracellular matrix was visualized by von Koss staining, which served as an indicator of osteogenic differentiation.

To study adipogenic differentiation, MSCs were seeded into 6-well culture plates (2.5 × 10^5^ cells/well) and cultured in a medium containing 90% DMEM-LG (Sigma-Aldrich, St. Louis, MO, USA), 10% inactivated FBS (Sigma-Aldrich, St. Louis, MO, USA), 100 U/mL penicillin, and 100 μg/mL streptomycin (Sigma-Aldrich, St. Louis, MO, USA), 0.5 μM hydrocortisone (Sigma-Aldrich, St. Louis, MO, USA), 60 μM indomethacin (Sigma-Aldrich, St. Louis, MO, USA), 1 μg/mL insulin (Sigma-Aldrich, St. Louis, MO, USA USA), 0.5 μM isobutyl-methylxanthine (Sigma-Aldrich, St. Louis, MO, USA, USA). The medium was changed every 4 days. The complete cultivation cycle was 21 days. Adipogenic differentiation was assessed by the intracellular accumulation of lipids stained with 0.5% Oil Red O (Sigma-Aldrich, St. Louis, MO, USA).

Granular cultures of MSCs for the study of chondrogenesis were prepared from 4 × 10^5^ cells placed in 15 mL polypropylene tubes (Techno Plastic Products AG, Trasadingen, Switzerland) in a medium of the following composition: 95% DMEM-HG (Sigma-Aldrich, St. Louis, MO, USA), 1% ITS+3 Liquid Media Supplement (Sigma-Aldrich, St. Louis, MO, USA), 1% antibiotic solution (penicillin/streptomycin 100 U/mL and 100 μg/mL (Sigma-Aldrich, St. Louis, MO, USA), 100 μg/mL pyruvate soda (Sigma-Aldrich, St. Louis, MO, USA), 50 μg/mL L-ascorbic acid 2-phosphate (AsAP, Sigma-Aldrich, St. Louis, MO, USA), 40 μg/mL L-proline (Sigma-Aldrich, St. Louis, MO, USA), 0.1 μM dexamethasone (Sigma-Aldrich, St. Louis, MO, USA), and 10 ng/mL recombinant human TGF-β1 (Sigma-Aldrich, St. Louis, MO, USA). The cells were cultured for 14 days; the medium was changed every 3 days. On day 14, the cell granules were removed, fixed in 10% formalin solution for 24 h, and embedded in paraffin. To detect sulfated proteoglycans, sections of granules were stained with toluidine blue.

Fibroblast differentiation of MSCs was studied in the following medium: 90% DMEM-HG (Sigma-Aldrich, St. Louis, MO, USA) with fibroblast growth factor 2 μg/mL (Sigma-Aldrich, St. Louis, MO, USA) supplemented with 10% inactivated FBS (Sigma-Aldrich, St. Louis, MO, USA), 280 μg/mL L-glutamine (Sigma-Aldrich, St. Louis, MO, USA), 50 μg/mL gentamicin (Serva, Heidelberg, Germany), 10^−8^ M dexamethasone (Sigma-Aldrich, St. Louis, MO, USA). The cell concentration was 6 × 10^5^ cells in 2 mL. On the 18th day cell cultures were fixed in 4% formaldehyde (Pancreas, San Jose, CA, USA) and stained with the main May-Grunwald stain [42].

#### 4.9.4. Formation of Fibroblast Colonies by CD45^‒^ Cells

After the study of the CD45 marker, the final concentration of the adherent cell elements of the lungs was adjusted to 0.5 × 10^6^/mL with a liquid nutrient medium of the following composition: 80% DMEM medium (Sigma-Aldrich, St. Louis, MO, USA), 20% FBS (Sigma-Aldrich, St. Louis, MO, USA), 280 mg/L l-glutamine (Sigma-Aldrich, St. Louis, MO, USA), 50 mg/L gentamicin (Sigma-Aldrich, St. Louis, MO, USA), 25 ng/mL fibroblast growth factor (Sigma-Aldrich, St. Louis, MO, USA). Then, 0.5 mL of the prepared suspension was placed in plastic 24-well plates (Corning, New York, NY, USA) and incubated in a CO_2_ incubator at 37 °C, 5% CO_2_, and 100% air humidity for 7 days. After incubation under an inverted microscope, the number of colonies was counted. Fibroblast colonies (CFU-F) meant cell aggregates containing more than 50 cells of fibroblast morphology [42].

### 4.10. Statistical Analysis

Statistical analysis was performed using SPSS (version 15.0, SPSS Inc., Chicago, IL, USA). Data were analyzed and presented as means ± standard error of the mean. Statistical significance was evaluated by Student’s t-test (for parametric data), or Mann–Whitney test (for nonparametric data) when appropriate. A *p*-value of less than 0.05 (by two-tailed testing) was considered an indicator of statistical significance.

## 5. Conclusions

Our study showed that poloxamer-hyaluronidase, created using electron beam synthesis, has anti-inflammatory and antifibrotic effects in animals with pulmonary fibrosis. The positive effect of treatment with pHD is significantly superior to that of native hyaluronidase, which can be explained by the resistance of poloxamer-hyaluronidase to the destructive action of various factors. The anti-inflammatory and antifibrotic effects, polytherapy with spiperone and pHD is superior to monotherapy with pHD and spiperone. We observed regeneration of endothelial and epithelial cells in this case.

Thus, sequential spiperone administration and destruction of hyaluronic acid by pHD may be a promising approach for the treatment of patients with IPF.

## Figures and Tables

**Figure 1 ijms-22-05599-f001:**
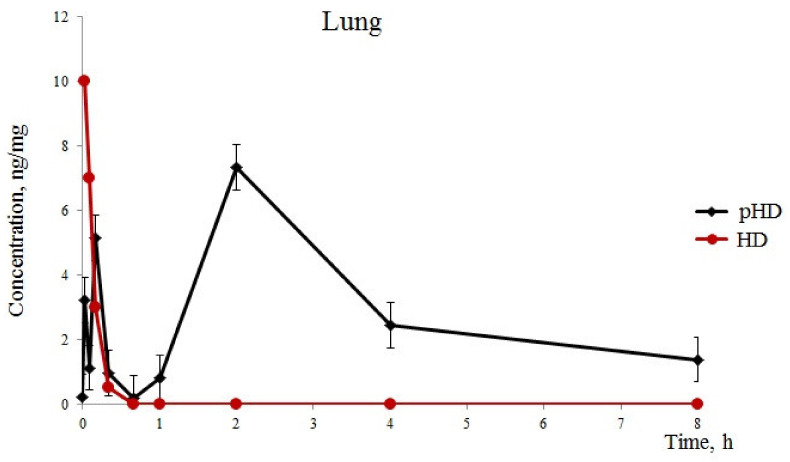
Hyaluronidase levels (standard error of the mean) in the lung. Pharmacokinetics of native hyaluronidase (HD) and poloxamer-hyaluronidase (pHD) in the lung tissue of mice after a single intranasal administration at a dose of 40 U/kg.

**Figure 2 ijms-22-05599-f002:**
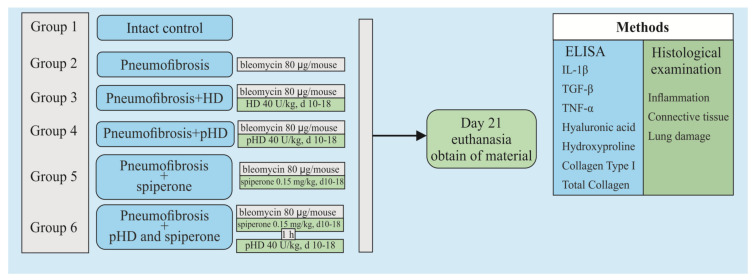
Graphical scheme of the protocol for inducing pulmonary fibrosis by bleomycin and the protocol for studying pHD. Groups: Intact control—Group 1; mice with bleomycin-induced pneumofibrosis—Group 2; mice with bleomycin-induced pneumofibrosis treated HD—Group 3; mice with bleomycin-induced pneumofibrosis treated pHD—Group 4; mice with bleomycin-induced pneumofibrosis treated Spiperone—Group 5; mice with bleomycin-induced pneumofibrosis treated Spiperone+pHD—Group 6.

**Figure 3 ijms-22-05599-f003:**
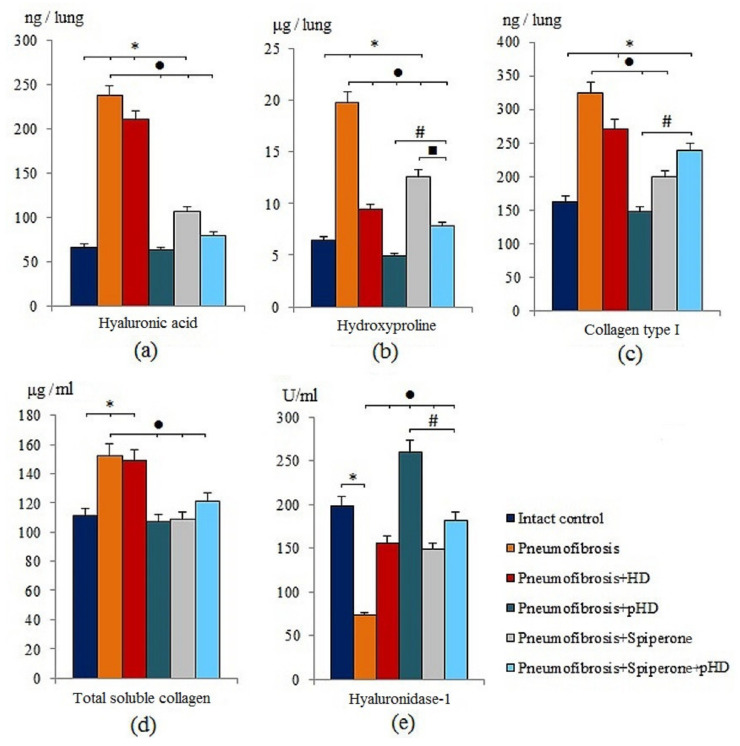
The levels of hyaluronic acid (**a**), hydroxyproline (**b**), and collagen type I (**c**), total soluble collagen (**d**), and hyaluronidase 1 (**e**) in homogenate of whole right lung lobes received from male C57BL/6 mice (d21). Groups: Intact control; mice with bleomycin-induced pneumofibrosis (Pneumofibrosis); mice with pneumofibrosis treated HD (Pneumofibrosis+HD); mice with pneumofibrosis treated pHD (Pneumofibrosis+pHD); mice with pneumofibrosis treated Spiperone (Pneumofibrosis+Spiperone); mice with pneumofibrosis treated Spiperone+pHD (Pneumofibrosis+Spiperone+pHD). *—*p* < 0.05 significance of difference compared with intact control group; ● *p* < 0.05 significance of difference compared with the Pneumofibrosis group; #—*p* < 0.05 significance of difference compared with the Pneumofibrosis+pHD group; ■—*p* < 0.05 significance of difference compared with the Pneumofibrosis+Spiperone group.

**Figure 4 ijms-22-05599-f004:**
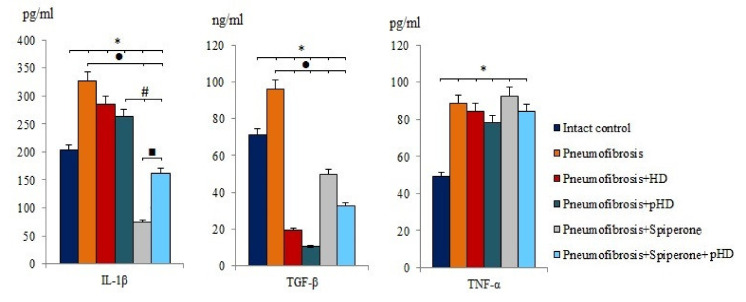
The levels of IL-1β, TGF-β, and TNF-α in lung homogenate received from male C57BL/6 mice (d21). Groups: Intact control; mice with bleomycin-induced pneumofibrosis (Pneumofibrosis); mice with pneumofibrosis treated HD (Pneumofibrosis+HD); mice with pneumofibrosis treated pHD (Pneumofibrosis+pHD); mice with pneumofibrosis treated Spiperone (Pneumofibrosis+Spiperone); mice with pneumofibrosis treated Spiperone+pHD (Pneumofibrosis+Spiperone+pHD). *— *p* < 0.05 significance of difference compared with intact control group; ● *p* < 0.05 significance of difference compared with the Pneumofibrosis group; #— *p* < 0.05 significance of difference compared with the Pneumofibrosis+pHD group; ■—*p* < 0.05 significance of difference compared with the Pneumofibrosis+Spiperone group.

**Figure 5 ijms-22-05599-f005:**
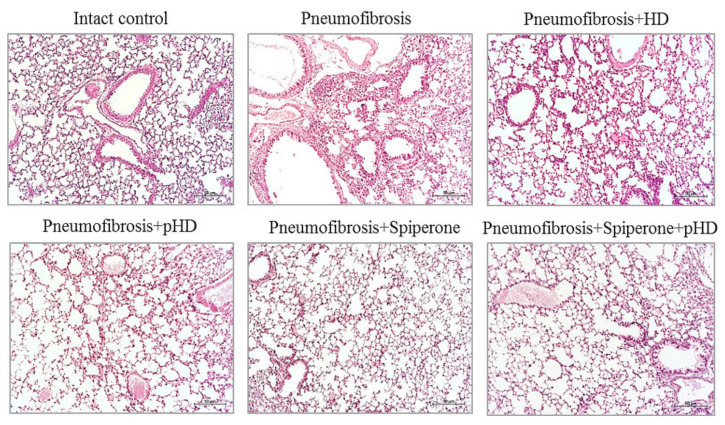
Photomicrographs of left lung sections (middle pulmonary field) obtained from male C57BL/6 mice on d21. Groups: Intact control; mice with bleomycin-induced pneumofibrosis (Pneumofibrosis); mice with bleomycin-induced pneumofibrosis treated HD (Pneumofibrosis+HD); mice with bleomycin-induced pneumofibrosis treated pHD (Pneumofibrosis+pHD); mice with bleomycin-induced pneumofibrosis treated Spiperone (Pneumofibrosis+Spiperone); mice with bleomycin-induced pneumofibrosis treated Spiperone+pHD (Pneumofibrosis+Spiperone+pHD). Tissues stained with hematoxylin-eosin, scale bar 50 μm.

**Figure 6 ijms-22-05599-f006:**
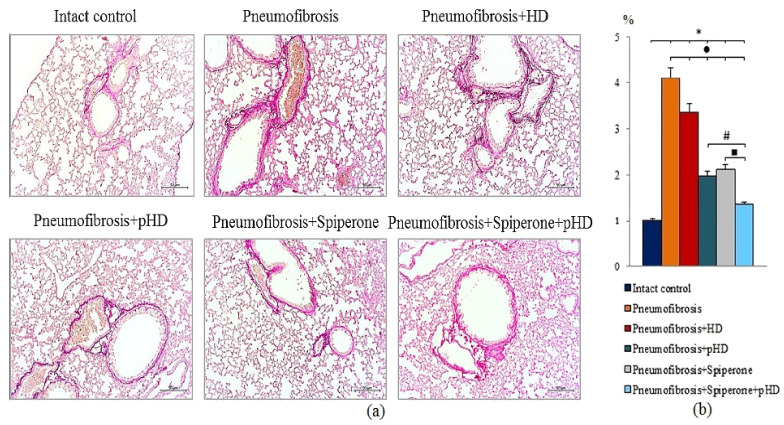
Photomicrographs of left lung sections (middle pulmonary field) obtained from male C57BL/6 mice on d21 (**a**). Groups: Intact control; mice with bleomycin-induced pneumofibrosis (Pneumofibrosis); mice with bleomycin-induced pneumofibrosis treated HD (Pneumofibrosis+HD); mice with bleomycin-induced pneumofibrosis treated pHD (Pneumofibrosis+pHD); mice with bleomycin-induced pneumofibrosis treated Spiperone (Pneumofibrosis+Spiperone); mice with bleomycin-induced pneumofibrosis treated Spiperone+pHD (Pneumofibrosis+Spiperone+pHD). Tissues stained by Van Gieson, scale bar 50 μm. Content of the connective tissue in the lungs of C57BL/6 mice with bleomycin-induced pneumofibrosis on d21 (**b**). Content of the connective tissue in the lungs was expressed as a percentage of the lung tissue area. *—*p* < 0.05 significance of difference compared with intact control group; ● *p* < 0.05 significance of difference compared with the Pneumofibrosis group; #—*p* < 0.05 significance of difference compared with the Pneumofibrosis+pHD group; ■—*p* < 0.05 significance of difference compared with the Pneumofibrosis+Spiperone group.

**Figure 7 ijms-22-05599-f007:**
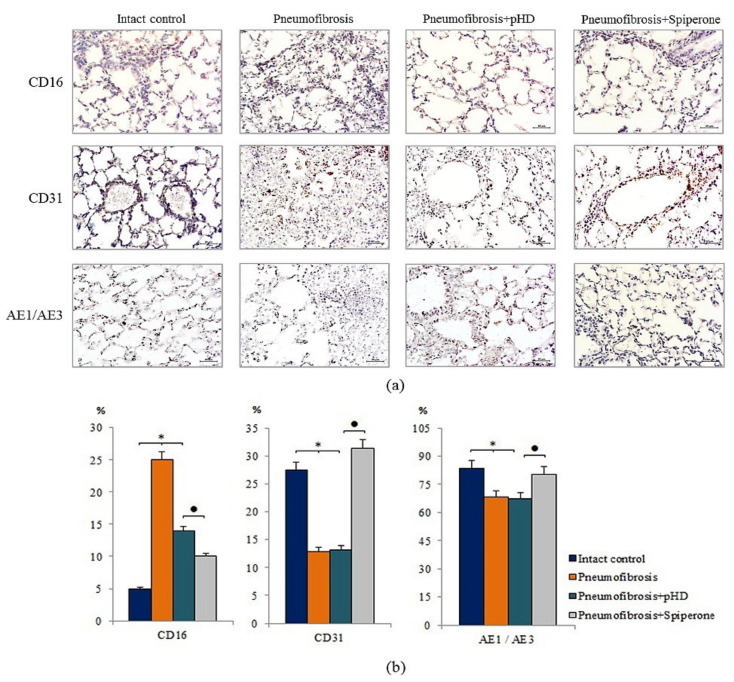
Micrographs of lung sections (middle pulmonary field) obtained from male C57BL/6 mice. Immunohistochemical staining (brown) for specific cell markers: CD16, CD31, and pan-cytokeratin (AE1/AE3) in the lungs of C57BL/6 mice (**a**). Scale bar 10 μm. Relative percentages of cells expressing CD16, CD31, and pan-cytokeratin (AE1/AE3) in the lungs of control mice (Intact control), mice with bleomycin-induced pneumofibrosis (Pneumofibrosis); mice with bleomycin-induced pneumofibrosis treated pHD (Pneumofibrosis+pHD); mice with bleomycin-induced pneumofibrosis treated Spiperone (Pneumofibrosis+Spiperone) assessed by immunohistochemical on d21 (**b**). *—*p* < 0.05 significance of difference compared with intact control group; ● *p* < 0.05 significance of difference compared with the Pneumofibrosis+pHD group.

**Figure 8 ijms-22-05599-f008:**
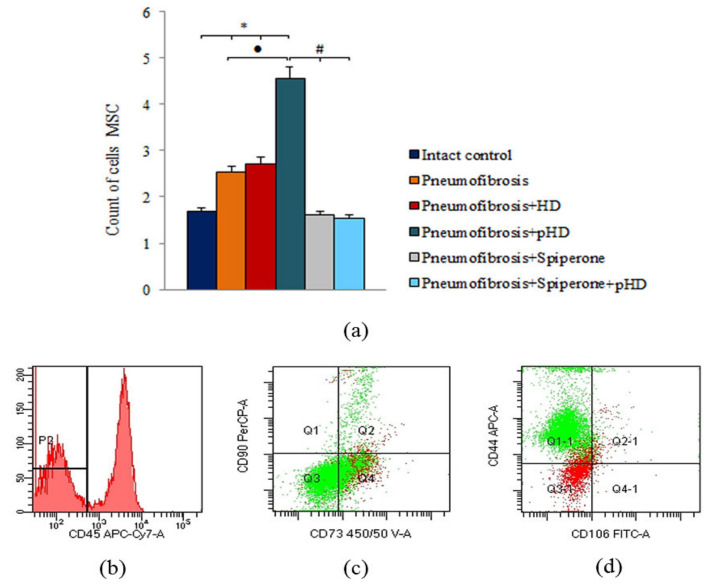
Characterization of MSCs cell population isolated from lung of male C57BL/6 mice on d21. Cells were analyzed by flow cytometry using antibodies for mouse CD45, CD44, CD73, CD90 and CD106. (**a**) Quantification showing mean values from three independent experiments. (**b**–**d**) Representative dot plots. Groups: Intact control; mice with bleomycin-induced pneumofibrosis (Pneumofibrosis); mice with bleomycin-induced pneumofibrosis treated HD (Pneumofibrosis+HD); mice with bleomycin-induced pneumofibrosis treated pHD (Pneumofibrosis+pHD); mice with bleomycin-induced pneumofibrosis treated Spiperone (Pneumofibrosis+Spiperone); mice with bleomycin-induced pneumofibrosis treated Spiperone+pHD (Pneumofibrosis+Spiperone+pHD). *— *p* < 0.05 significance of difference compared with intact control group; ● *p* < 0.05 significance of difference compared with the Pneumofibrosis group; #— *p* < 0.05 significance of difference compared with the Pneumofibrosis+pHD group.

**Figure 9 ijms-22-05599-f009:**
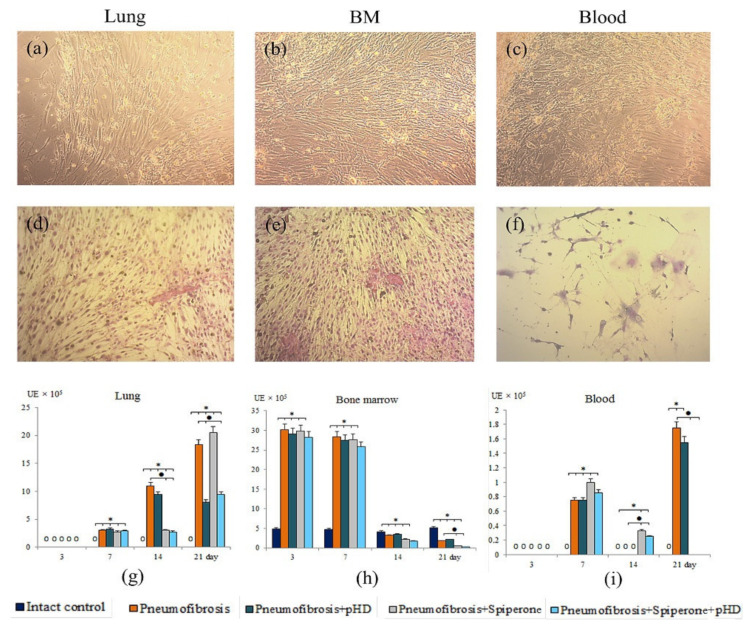
Photomicrographs of culture of fibroblast progenitor cells isolated from lung (**a**,**d**); bone marrow (**b**,**e**); and blood (**c**,**f**) male C57BL/6 mice of intact control. Native culture (**a**–**c**) and culture stained with hematoxylin-eosin (**d**–**f**). 100× Clonal activity of CD45^-^ fibroblast progenitor cells (CFU-F, ×10^5^ adherent mononuclear cells) isolated from lung (**g**), bone marrow (**h**), and blood (**i**) of male C57BL/6 mice after bleomycin introduction. Groups: Intact control; mice with bleomycin-induced pneumofibrosis (Pneumofibrosis); mice with bleomycin-induced pneumofibrosis treated pHD (Pneumofibrosis+pHD); mice with bleomycin-induced pneumofibrosis treated Spiperone (Pneumofibrosis+Spiperone); mice with bleomycin-induced pneumofibrosis treated Spiperone+pHD (Pneumofibrosis+Spiperone+pHD). *—*p* < 0.05 significance of difference compared with intact control group; ● *p* < 0.05 significance of difference compared with the Pneumofibrosis group.

**Figure 10 ijms-22-05599-f010:**
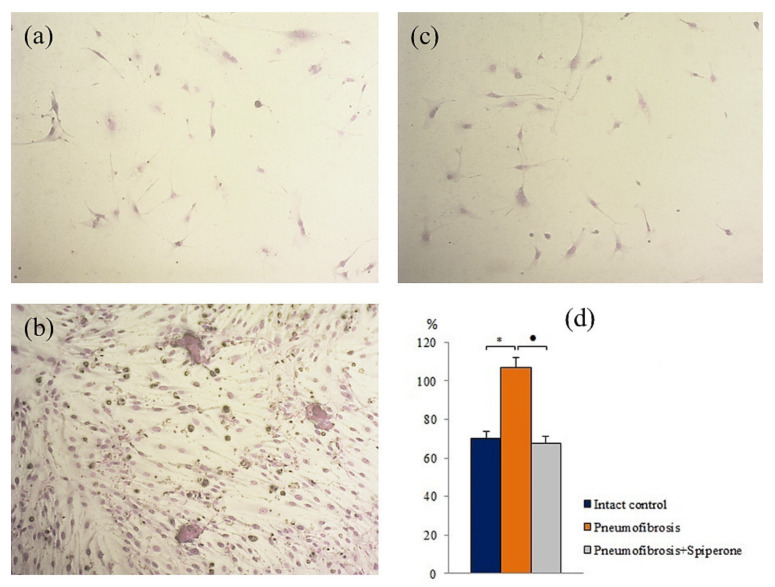
Photomicrographs of fibroblast differentiation of of primary culture adherent cells isolated from the lungs of C57BL/6 mice from groups: Intact control (**a**); mice with bleomycin-induced pneumofibrosis (Pneumofibrosis) (**b**); mice with bleomycin-induced pneumofibrosis treated Spiperone (Pneumofibrosis+Spiperone) 100× (**c**). The presence of fibroblasts in culture was confirmed by staining blue by May-Grunwald. Fibroblast differentiation (**d**). The ordinate axis: fibroblasts (% of total number of mononuclears) (**d**). *—*p* < 0.05 significance of difference compared with intact control group; ● *p* < 0.05 significance of difference compared with the Pneumofibrosis group.

**Figure 11 ijms-22-05599-f011:**
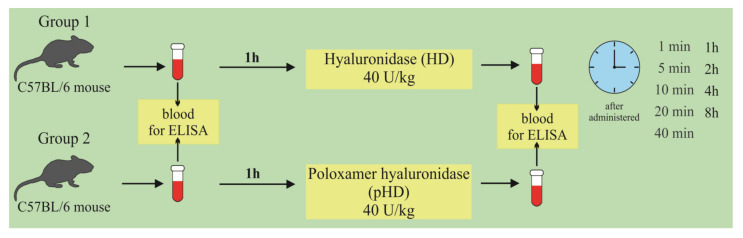
Graphical scheme of the protocol for studying the pharmacokinetics of native hyaluronidase and poloxamer-hyaluronidase in the lung tissue of mice after a single intranasal administration at a dose of 40 U/kg.

**Figure 12 ijms-22-05599-f012:**
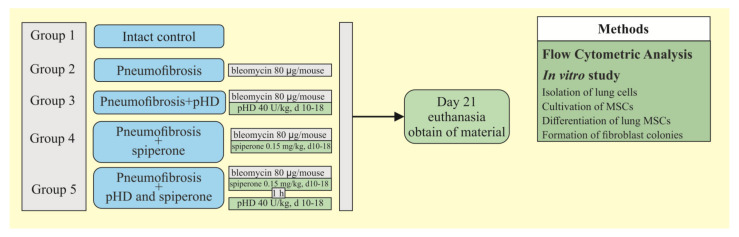
Graphical scheme of the protocol for studying the effect of spiperone and pHD on MSCs and fibroblast progenitor cells.

## Data Availability

Not applicable.

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
