# Peer review of "Micellar Hyaluronidase and Spiperone as a Potential Treatment for Pulmonary Fibrosis"

_ijms, 2021, doi:10.3390/ijms22115599_

Round 1

Reviewer 1 Report

The manuscript entitled ‘Prospects for Modulation of the Lung Hyaluronic Matrix by Micellar Hyaluronidase and Blockade of D2 Dopamine Receptors with Spiperone for the Treatment of Pulmonary Fibrosis’ by Skurikhin et al, is an analysis of the effect of hyaluronidase, poloxamer hyaluronidase and spiperone on bleomycin induced fibrosis in mouse model and in vitro effects on isolated cells. To this end, the authors investigated changes in vivo like inflammation and fibrosis and in vitro changes in progenitor cell mobilization. The data on the use of poloxamer hyaluronidase and spiperone is potentially an advance forward and also interesting. However, detailed investigation seems lacking in the manuscript.

Major comments

  1. The authors used spiperone in fibrotic conditions and with data proposed the involvement of D2 dopamine receptors. The link of spiperone to Drd receptors is known, however, to specifically point to their role, the authors are required to show that spiperone is not effective in conditions lacking the receptors. Otherwise, the Drd link is only a possibility and therefore removed from the title and text rephrased accordingly. There is also the issue of the amount of expression of Drd and the cell type of expression. Do the authors have data in what cells Drd2 is expressed in lungs and if the spiperone is acting specifically through cells expressing Drd2?
  2. The authors mention the use of SircolTM assay to measure ‘total soluble collagen’, but mention ‘total collagen’ in figure panels. Why the change? The amount of collagen per lung does not seem to match as reported in the literature. If it was the right lung, then is the whole right lung or one lobe? Specificity is missing in the figure panels.
  3. For figure panels with histological analysis, do the authors have the complete lung area e.g. the left lungs? Larger areas with zoom-in will be easier to appreciate the differences. Also, the pictures have different backgrounds e.g. some with yellowish background.
  4. It is not entirely clear why the authors studied the multilineage (replace multilinear in line 292) differentiation of MSCs. How much are they contributing to fibrosis in these conditions? It will be more important to see the effect of differentiation of fibroblast and myofibroblasts. It will be necessary to see myofibroblast markers and changes with the drugs at least in vitro conditions.

Minor comments

  1. The manuscript will highly improve if checked by a native English speaker.
  2. If mentioning emails, then check that all author emails are mentioned along with correct address.
  3. Line 262, replace cultural with cell culture.

Author Response

We thank the reviewer for their time and valuable comments. We have now revised the manuscript according to the suggestions. Please find below the reviewers’ comments and our responses. All changes have been included in the revised manuscript.

 Major comments

  1. The authors used spiperone in fibrotic conditions and with data proposed the involvement of D2 dopamine receptors. The link of spiperone to Drd receptors is known, however, to specifically point to their role, the authors are required to show that spiperone is not effective in conditions lacking the receptors. Otherwise, the Drd link is only a possibility and therefore removed from the title and text rephrased accordingly. There is also the issue of the amount of expression of Drd and the cell type of expression. Do the authors have data in what cells Drd2 is expressed in lungs and if the spiperone is acting specifically through cells expressing Drd2?

 We fully agree with the reviewer that we did not show that spiperone is not effective in conditions lacking the receptors. We used spiperone as a dopamine D2 receptor antagonist. While spiperone displays antipsychotic activity, it is not used as an antipsychotic. Rather it is widely used as a pharmacological tool for studying neurotransmitter receptors. Spiperone played a key role in characterizing the serotonin 5-HT1 and 5-HT2 receptors. It is known dopamine receptors are abundant in the CNS, and also found outside the brain. Dopamine 1 Receptor-like receptors are expressed on fibroblasts, lung mesenchyme [1], a small percentage of nerve fibers contained in pulmonary nerve trunks [4], while D2R-like receptors are mainly expressed in prejunctional sympathetic nerve endings [2], in pulmonary trunks [3]. However, the lung tissue is rich in dopamine and dopamine receptors (D1-4 type) [5].

We proposed that the effects of spiperone can be associated with D2 dopamine receptors. We now changed the title and text rephrased accordingly. We plan to investigate the effect of spiperone on the amount of expression of DR and the cell type of expression in our next study.

References.

  1. Haak, A. J., Kostallari, E., Sicard, D., Ligresti, G., Choi, K. M., Caporarello, N., Jones, D. L., Tan, Q., Meridew, J., Diaz Espinosa, A. M., Aravamudhan, A., Maiers, J. L., Britt, R. D., … Tschumperlin, D. J. Selective YAP/TAZ inhibition in fibroblasts via dopamine receptor D1 agonism reverses fibrosis. Science translational medicine, 2019. 11(516), eaau6296. https://doi.org/10.1126/scitranslmed.aau6296
  2. Beaulieu JM, Gainetdinov RR. The physiology, signaling, and pharmacology of dopamine receptors. Pharmacol Rev. 2011 Mar;63(1):182-217. doi: 10.1124/pr.110.002642.
  3. Amenta F, Ricci A, Tayebati SK, Zaccheo D. The peripheral dopaminergic system: morphological analysis, functional and clinical applications. Ital J Anat Embryol. 2002 Jul-Sep;107(3):145-67. PMID: 12437142.
  4. Feng, Y., & Lu, Y. Immunomodulatory Effects of Dopamine in Inflammatory Diseases. Frontiers in immunology, 2021. 12, 663102. doi.org/10.3389/fimmu.2021.663102
  5. Amenta F, Ricci A, Tayebati SK, Zaccheo D. The peripheral dopaminergic system: morphological analysis, functional and clinical applications. Ital. J. Anat. Embryol. -2002. - Vol. 107. - P. 145-167.

 2. The authors mention the use of SircolTM assay to measure ‘total soluble collagen’, but mention ‘total collagen’ in figure panels. Why the change? The amount of collagen per lung does not seem to match as reported in the literature. If it was the right lung, then is the whole right lung or one lobe? Specificity is missing in the figure panels.

 We thank the reviewer for spotting this inaccuracy. We now changed Figure 3 and the information in the text of Materials and Methods. The amount of collagen per whole right lung was measured. Results are shown as µg collagen per ml. The amount of collagen per lung is in accordance with the amounts reported in the literature [1].  

  1. Lin An, Li-Ying Peng, Ning-Yuan Sun, Yi-Lin Yang, Xiao-Wei Zhang, Bin Li, Bao-Lin Liu, Ping Li, and Jun Chen.Antioxidants & Redox Signaling.May 2019.1831-1848.http://doi.org/10.1089/ars.2018.7569

 3. For figure panels with histological analysis, do the authors have the complete lung area e.g. the left lungs? Larger areas with zoom-in will be easier to appreciate the differences. Also, the pictures have different backgrounds e.g. some with yellowish background.

 We now changed figure panels with histological analysis and added information.

 4. It is not entirely clear why the authors studied the multilineage (replace multilinear in line 292) differentiation of MSCs. How much are they contributing to fibrosis in these conditions? It will be more important to see the effect of differentiation of fibroblast and myofibroblasts. It will be necessary to see myofibroblast markers and changes with the drugs at least in vitro conditions.

 We now have corrected mistakes.

We studied mesenchymal stem cells (MSCs) surface markers. It is known MSCs are multipotent cells that are capable of osteogenic, chondrogenic, adipogenic, and fibrogenic differentiation. Therefore we studied the multilineage differentiation of MSCs further. We wanted to receive proof that we studied really MSC and how multilineage differentiation was changed at the pulmonary fibrosis. Fibrogenic differentiation is more important in pulmonary fibrosis because fibrocytes participate in collagen deposition at pulmonary fibrosis. We plan to study the effect of drugs on fibroblast and myofibroblasts differentiation in our next study.

We have previously obtained data the demonstrated the anti-inflammatory and anti-fibrotic effects of spiperone. We hypothesized that co-administration of spiperone and poloxamer hyaluronidase could have higher therapeutic efficacy compared to monotherapy with each respective compound.

 Minor comments

  1. The manuscript will highly improve if checked by a native English speaker.

We now have corrected it.

2. If mentioning emails, then check that all author emails are mentioned along with correct address.

We now have corrected it. We now have changed author emails for Pavel Madonov - [email protected], and Andrey Artamonov - [email protected]

3. Line 262, replace cultural with cell culture.

We now have corrected it.

Reviewer 2 Report

In this MS, authors have investigated the beneficial effect go Hyaluronic acid and their modified drug conjugate in presence and absence of Spiperone in the animal model of Pulmonary Fibrosis. Authors have demonstrated the beneficial effect of pHD and Spiperone as a promising approach for the treatment of IPF. Here are the comments - 

Major

  1. Authors have used 40U of HD in this paper and previously 100 HD and in their another publication, they have used 40U/kg, which corresponds to 1.2 U of HD (25927611). This shows a high level of variability in their own work between the years. Even if the source of HD is different batch, such a variability raises an attention, which authors need to discuss in the introduction or in discussion. Authors need to discuss their PlosOne publication in the introduction, where they talk about ref 14. As a proof of concept, authors should produce a pilot experiment comparing the effect of pegylated HD+Spiperone and pHD+Spiperone in Bleo animals.
  2. Fig 7 - Authors have not shown the IHC images of CD16, CD31 and AE1/AE3. Further, how did they quantify? Authors need to show the images and they should do the quantification by PCR for western for the above markers.
  3. The justification of an increase in MSC needs to be further validated by assessing the markers of EMT (Epithelial Mesenchymal Markers) in lung tissue for all the 6 groups and add those data to Fig 8. Also, please assess the expression of lung remodeling markers such as MMPs. Please show the flow raw images of Flow in Fig 8 or in supplements. 
  4. Authors need to show the images of specific staining for the differentiation of MSCs, such as Oil Red O, Alcian Blue and etc.

Minor

  1. Authors strongly claim that the combination would be a promising approach for the treatment of IPF, while only prevention study has been performed. Authors should do a reveal or treatment protocol and make those claims or tone down the language to state that this 'suggest a beneficial action' or something like 'the combination improves or attenuates the PF'
  2. Reference 6 - Authors claim this as laboratory animals. Authors should mention this as Bleo animals. 
  3. Fig 1 - Please make the scatter plot on the line differentiated - one as open and one as close circle or some other shape
  4. It is difficult to differentiate the lung tissue remodeling based on the images shown in Fig 6, while the data claims so. Authors need to show the image at higher magnification - 200 or 400 or 650X and explicitly show the effect of tissue remodeling in Bleo animals

Author Response

We thank the reviewer for their time and valuable comments. We have now revised the manuscript according to the suggestions. Please find below the reviewers’ comments and our responses. All changes have been included in the revised version.

Major

  1. Authors have used 40U of HD in this paper and previously 100 HD and in their another publication, they have used 40U/kg, which corresponds to 1.2 U of HD (25927611). This shows a high level of variability in their own work between the years. Even if the source of HD is different batch, such a variability raises an attention, which authors need to discuss in the introduction or in discussion. Authors need to discuss their PlosOne publication in the introduction, where they talk about ref 14. As a proof of concept, authors should produce a pilot experiment comparing the effect of pegylated HD+Spiperone and pHD+Spiperone in Bleo animals.

Thank you for your question. Our manuscript includes information about the mechanism influence of poloxamer hyaluronidase and spiperone on bleomycin-induced fibrosis. We have used the maximum effective dose to investigate the effects of poloxamer hyaluronidase (40 U/kg in a volume 8 μL buffer/mouse) on fibrosis. Therefore, we have chosen an equivalent dose of the native hyaluronidase (40 U/kg in a volume of 8 μL/mouse) preparation in our study. The dose of pHD and HD is identical in this publication.

In our PlosOne publication, we showed that treatment with pegHYAL (8 U/18 μL buffer/mouse) and HYAL (8 U/18 μL buffer/mouse) with spiperone reduced the bleomycin-induced fibrotic response in the lung parenchyma. The dose of pegylated HD and dose HD is identical in this publication too.

In our publication, the poloxamer hyaluronidase is a new formulation of hyaluronidase. We have studied the pharmacological activity of a new drug. We did not study and compare the effect of pegylated HD+Spiperone and poloxamer HD+Spiperone in animals with bleomycin-induced fibrosis. These are different drugs based on hyaluronidase.

2. Fig 7 - Authors have not shown the IHC images of CD16, CD31 and AE1/AE3. Further, how did they quantify? Authors need to show the images and they should do the quantification by PCR for western for the above markers.

We now have corrected it. We now added information in the Fig 7.

3. The justification of an increase in MSC needs to be further validated by assessing the markers of EMT (Epithelial Mesenchymal Markers) in lung tissue for all the 6 groups and add those data to Fig 8. Also, please assess the expression of lung remodeling markers such as MMPs. Please show the flow raw images of Flow in Fig 8 or in supplements. 

We fully agree with Reviewer that needs to assess the markers of EMT (Epithelial Mesenchymal Markers) in lung tissue.

We studied MSCs surface markers. CD45 negative cells extracted from lungs of mice C57BL/6 of control and experimental groups were positive for the markers CD44, CD73, CD90, and CD106. We wanted to receive proof that we studied really MSC, therefore we studied multilineage differentiation of MSC. Additionally, we studied lung tissue by histological and immunohistochemical examination, we evaluated the level of IL-1β, TGF-β, TNF-α, hyaluronic acid, hydroxyproline, collagen type I, and total soluble collagen.  We aimed to evaluate the effect of poloxamer hyaluronidase on bleomycin-induced alveolar damage, inflammation, and lung fibrosis in C57BL/6 mice in comparison with native hyaluronidase. Taken together, our results show sequential spiperone administration and destruction of hyaluronic acid by pHD may suggest a beneficial action of these drugs on IPF.

We understand our scientific work has limitations. This is part of a big experimental study. 

It is known lung fibrosis has long been categorized as a type II EMT event, but the cellular networks that contribute to tissue scarring have not been well characterized in humans. Therefore we plan to continue our work and show the markers of EMT in lung tissue in next our paper.

We now have corrected figure 8.

4. Authors need to show the images of specific staining for the differentiation of MSCs, such as Oil Red O, Alcian Blue and etc.

We now have corrected it. Mesenchymal stem cells are multipotent cells that are capable of osteogenic, chondrogenic, adipogenic, and fibrogenic differentiation. The increase of fibrogenic differentiation is the main problem in pulmonary fibrosis. It is more important to see the effect of differentiation of fibroblast. Therefore we added specific staining for the fibroblastic differentiation of MSCs. in fig 10. We added the images of specific staining for the differentiation of MSCs, such as Oil Red O, Alcian Blue in Suppl.fig S1.

Minor

  1. Authors strongly claim that the combination would be a promising approach for the treatment of IPF, while only prevention study has been performed. Authors should do a reveal or treatment protocol and make those claims or tone down the language to state that this 'suggest a beneficial action' or something like 'the combination improves or attenuates the PF'

We now have corrected it.

2. Reference 6 - Authors claim this as laboratory animals. Authors should mention this as Bleo animals. 

Authors should mention this as Bleo animals. Various authors described laboratory animals differently. We described groups of laboratory animals of the standard method as in Reference 6 and other papers.

3. Fig 1 - Please make the scatter plot on the line differentiated - one as open and one as close circle or some other shape

We presented results of the concentration of hyaluronidase in the lungs of mice after a single intranasal administration of HD and pHD in healthy mice in figure 1. Based on these data, we revealed significantly greater stability of pHD in comparison with native HD with intranasal administration. We plan to present the results of pharmacokinetics and bioavailability of pHD in separate our manuscript. We presented our results as described in papers [1,2].

  1. Atkinson HC, Stanescu I, Frampton C, Salem II, Beasley CP, Robson R. Pharmacokinetics and Bioavailability of a Fixed-Dose Combination of Ibuprofen and Paracetamol after Intravenous and Oral Administration. Clin Drug Investig. 2015 Oct;35(10):625-32. doi: 10.1007/s40261-015-0320-8. 
  2. Pozharitskaya ON, Shikov AN, Obluchinskaya ED, Vuorela H. The Pharmacokinetics of Fucoidan after Topical Application to Rats. Mar Drugs. 2019 Dec 6;17(12):687. doi: 10.3390/md17120687. 

4. It is difficult to differentiate the lung tissue remodeling based on the images shown in Fig 6, while the data claims so. Authors need to show the image at higher magnification - 200 or 400 or 650X and explicitly show the effect of tissue remodeling in Bleo animals

We now have corrected it.

Round 2

Reviewer 1 Report

The manuscript has improved. Some issues still need to be addressed.

Minor comments

  1. Some figures have two different fonts. e.g. in Figure 1.
  2. Figure 6 has spelling mistake of Spiperone. It’s ‘Spiperon’ in figures.
  3. Figure 5 and 6 do not have similar histology of panel of pnemofibrosis + spiperone + pHD. Why in the Figure 6 the panel does not show clear alveolar structures. The quantification shows downregulation in this condition, but not clear in the panel. Can the authors show another representative image?
  4. Figure 7, the pictures are not clearly showing the staining. E.g. CD31 should be clearly visible in vessels. Here, in the pictures all looks the same color.
  5. Where are methods related to immunohistochemistry? Details of antibodies used in IHC and catalog numbers are missing.

Author Response

We thank the reviewer for their time and valuable comments. We have now revised the manuscript according to the suggestions. Please find below the reviewers’ comments and our responses. All changes have been included in the revised manuscript.

Minor comments

  1. Some figures have two different fonts. e.g. in Figure 1.

We now have corrected it.

2. Figure 6 has spelling mistake of Spiperone. It’s ‘Spiperon’ in figures.

We now have corrected mistakes.

3. Figure 5 and 6 do not have similar histology of panel of pnemofibrosis + spiperone + pHD. Why in the Figure 6 the panel does not show clear alveolar structures. The quantification shows downregulation in this condition, but not clear in the panel. Can the authors show another representative image?

We now have corrected Figure 6.

Figure 5 shows photomicrographs of left lung sections where tissues stained with hematoxylin-eosin. Staining of lung preparations with hematoxylin and eosin made it possible to evaluate inflammatory cells in the lung.

Figure 6 shows photomicrographs of left lung sections where tissues stained by Van Gieson. We evaluated the content of the connective tissue in the lungs of C57BL/6 mice. We added information in the supplement. Figures S1 and S2 show other representative images of the lung.

4. Figure 7, the pictures are not clearly showing the staining. E.g. CD31 should be clearly visible in vessels. Here, in the pictures all looks the same color.

We now have corrected Figure 7.

5. Where are methods related to immunohistochemistry? Details of antibodies used in IHC and catalog numbers are missing.

We now have corrected it. We now added information in Material and Methods.

Reviewer 2 Report

Authors have attempted to address the raised concerns. Regarding the laboratory animal comment - It is still highly misleading! The reference 6 is used to talk for Bleo animals in the paragraph 2 of the introduction. With in few lines, the same reference is discussed as laboratory animals! Authors should fix this!

Author Response

We thank the reviewer for their time and valuable comments.

We now added information about Bleo animals in paragraph 2 of the introduction.

Round 3

Reviewer 1 Report

The manuscript has improved. Still some minor issues:

  1. Line 632, what is X (chi)? It is not mentioned in the text. Please mention X in brackets with content / percentage of collagen fibres.
  2. In the y axis in the Figure 6, what is the % of area of lung tissue? Shouldn’t it be % of area of connective tissue, as in the figure legend?
  3. IHC in the Figure 7 are still not convincing. The panel of CD31 and AE1, AE3 related to pneumofibrosis, seem to be non-fibrotic areas. While that of Cd16 seems to be of the fibrotic area. In the methods, please clearly describe how the quantification were performed so that the data can be reproduced. Just writing that it was done in ImageJ is not enough. The authors mentioned that they counted cells. In how many pictures?

Author Response

We thank the reviewer for their time and valuable comments.

  1. Line 632, what is X (chi)? It is not mentioned in the text. Please mention X in brackets with content / percentage of collagen fibres.

We now have corrected it.

  1. In the y axis in the Figure 6, what is the % of area of lung tissue? Shouldn’t it be % of area of connective tissue, as in the figure legend?

We now have corrected mistakes.

  1. IHC in the Figure 7 are still not convincing. The panel of CD31 and AE1, AE3 related to pneumofibrosis, seem to be non-fibrotic areas. While that of Cd16 seems to be of the fibrotic area. In the methods, please clearly describe how the quantification were performed so that the data can be reproduced. Just writing that it was done in ImageJ is not enough. The authors mentioned that they counted cells. In how many pictures?

We now have corrected Figure 7. We now added information in Material and Methods and reference.
